# Plasma microRNAs biomarkers in mild cognitive impairment among patients with type 2 diabetes mellitus

Iman I. Salama[1]*, Samia M. Sami[2], Ghada A. Abdellatif[1], Amira Mohsen[1], Hanaa Rasmy[3], Solaf Ahmed Kamel[3], Mona Hamed Ibrahim[3], Mona Mostafa[4], Walaa A. Fouad[1], Hala M. Raslan[4]

1 Community Medicine Department, National Research Centre, Cairo, Egypt, 2 Child Health Department, National Research Centre, Cairo, Egypt, 3 Clinical and Chemical Pathology Department, Centre of Excellence, National Research Centre, Cairo, Egypt, 4 Internal Medicine Department, National Research Centre, Cairo, Egypt

* salamaiman@yahoo.com

## Abstract

### Objectives

To assess the potential value of some miRNAs as diagnostic biomarkers for mild cognitive impairment (MCI) among patients with type2 diabetes mellitus (T2DM) and to identify other risk factors for MCI among them.

### Methods

This study enrolled 163 adults with T2DM using face to face interview. Cognitive function with its domains was assessed using Adenbrooke's Cognitive Examination III (ACE III). Lipid profile, glycated hemoglobin, and miR-128, miR-132, miR- 874, miR-134, miR-323, and miR-382 expressions, using quantitative real-time PCR, were assessed.

### Results

MCI was detected among 59/163 (36.2%) patients with T2DM. Plasma expression of miR-132 was significantly higher in T2DM patients with MCI compared to those without MCI and to normal cognitive healthy individuals (median = 2, 1.1 and 1.2 respectively, P < 0.05. Logistic regression analysis showed that higher miR-132 expression with adjusted odds ratio (AOR): 1.2 (95% CI 1.0–1.3), female gender (AOR:2.1; 95%CI 1.0–4.3), education below postgraduate (secondary and university education with AOR: 9.5 & 19.4 respectively) were the significant predicting factors for MCI among T2DM patients. Using ROC curve, miR-132 was the only assayed miRNA that significantly differentiates T2DM patients with MCI from those with normal cognition with 72.3% sensitivity, 56.2% specificity, and 63.8% accuracy (P < 0.05). Other studied miRNAs showed lower sensitivity and specificity for detecting MCI among studied T2DM participants.

**Data Availability Statement:** All data are fully available without restriction within the Supporting Information files.

**Funding:** This project is supported financially by the Science and Technology Development Fund (STDF), Egypt, Grant No: 15026, PI: Iman I Salama. STDF Grant No 4880 for capacity building for laboratory Equipment, PI: Hanaa Rasmy. Website: http://stdf.eg/. The funders had no role in study design, data collection and analysis, decision to publish, or preparation of the manuscript.

**Competing interests:** The authors have declared that no competing interests exist.

## Conclusion

MCI affects nearly one-third of adult patients with T2DM. A significantly over expression of miR-132 was detected among T2DM with MCI compared to those with normal cognition.

## Introduction

Type 2 diabetes mellitus (T2DM) is a major health problem all over the world and its incidence is rapidly increasing. Adverse effects of diabetes on cognitive function and memory disorders have been reported by researchers for a long time [1]. Diabetic patients are at higher risk for dementia, Alzheimer disease (AD), and vascular dementia than people with normal glucose tolerance [2]. The presentation of cognitive impairment ranges from subtle executive dysfunction to memory loss and overt dementia. Mild cognitive impairment (MCI) is of particular interest because of its influence on self-care and quality of life and its increased risk to progress to dementia including AD. Moreover, the prognosis of MCI among diabetics is worse than among non diabetics [2,3].

Glycemic control, hypoglycemia, dyslipidemia, micro and macrovascular complications have been reported as risk factors for cognitive dysfunction in T2DM but each appears to have limited effects [4]. On the other hand studies reported that intensive control of blood glucose, blood pressure, or cholesterol levels in T2DM is not beneficial in preventing cognitive decrements [5]. There are still substantial knowledge gaps on how the risk factors interact and affect cognition and the potential of the presence of genetic factors [6]. microRNAs (miRNAs) are widely distributed within the nervous system. They are key regulators of many biological functions such as neurogenesis, dendritic spine morphology, and synaptic plasticity. Moreover there is increasing evidence for their role in neurodegenerative disorders [7]. Many studies investigated the correlation of miRNAs dysregulation with AD. Butovsky and co-workers [8] observed the presence of miR-132 in all of the cell populations of the CNS with the strongest expression in the neurons, followed by astrocytes, oligodendrocytes, and finally, microglia. Moreover, miR-128, miR-132, and miR-874 paired with miR-491, while miR-134, miR-323, and miR-382 paired with miR-370 have been validated as sensitive biomarkers for detection of MCI [9]. The scientific interest in miRNAs come from their potential value as disease biomarkers and as a molecular target of disease therapies [10].

The present study aimed at assessing the potential value of miR-128, miR-132 and miR-874, miR-134, miR-323, and miR-382 as diagnostic biomarkers for MCI among a sample of Egyptian T2DM patients and the impact of the conventional risk factors on MCI among these patients.

## Materials and methods

This study was conducted along one year (from June 2016 to December 2017). It involved 163 T2DM patients, 65 males and 98 females; aged from 40 to 60 years from outpatient clinics of Medical Services Unit of the National Research Center (NRC). Patients were already diagnosed as T2DM and were receiving anti-diabetics drugs, either insulin or oral hypoglycemic drugs. All patients were literate and were able to complete the tests of cognitive function. T2DM patients with history of head trauma, stroke, transient ischemic attack, brain tumor, epilepsy, psychiatric disease, cardiac or liver failure, history of thyroid disease, visual or hearing disabilities, and acute or chronic infection were excluded. Inorder to compare the miRNAs

expressions among T2DM patients to normal cognitive non T2DM individuals, 109 non diabetic apparently healthy normal cognition individuals (age, sex, and education matched to the studied T2DM patients) were participated.

The study was conducted in accordance with the Code of Ethics of the World Medical Association (Declaration of Helsinki). Each participant provided an informed written consent after acknowledgement about the research. The study was approved by the Medical Research Ethical Committee of the NRC {registration number 15131}. A face to face interview was carried out where a closed ended questionnaire module was used to cover data on demographic characteristics and full medical history. Data included age, gender, level of education, and tobacco smoking. Detailed medical history of diabetes was taken from the patients including age at detecting T2DM, number and frequency of hypoglycemic events, type of treatment taken for diabetes (insulin or oral hypoglycemic drugs), history of hypertension, dyslipidemia, history of other diseases, and medications taken. Any complaint suggestive the presence of sensory peripheral neuropathy such as tingling, burning pain or numbness in hands and feet was recorded. All the studied participants were subjected to thorough clinical examination with anthropometric assessment (weight and height measurements) and body mass index (BMI) was calculated as weight in kg/height in $m^2$. Obesity was defined as BMI $\geq$ 30kg/$m^2$. Hypertension was defined as blood pressure above 140/90 or patients on anti-hypertensive medications. Peripheral sensory neuropathy was diagnosed by history and positive pinprick and/or filament test.

## Assessment of cognitive function with its domains

The case definition of MCI was based on the objective impairment in one or more cognitive domains. Adenbrooke's Cognitive Examination III (ACE III) test was chosen to assess global and specific cognitive domains. It is the most reliable and validated Arabic form cognitive scale and freely available to be applied. Objective MCI was considered if ACE III score is less than 88 Preservation of independence in daily functional abilities using SF36 quality of life was used to exclude dementia [11].

## Laboratory analysis

Venous blood samples after an overnight fast (12 hours) were withdrawn from all participants. Part of the blood sample was anticoagulated with EDTA for assessment of the glycated hemoglobin (HbA1c) (using Labona check™ HbA1c analyzer) and for measurement of selected miRNAs by real-time PCR. The other part of blood sample was left to clot and sera were separated immediately for analysis of lipid profile by Erba xl -300 Mannheim Gmbh Germany. Patients were diagnosed to have dyslipidemia if total cholesterol is above 200mg/dl and/or triglycerides above 150 mg/dl or patient on anti- hyperlipidemic drugs.

## miRNA gene expression

Blood samples were collected in EDTA-containing vacutainer. After 20 min centrifugation (2500×$g$), plasma was separated, aliquoted and stored at – 80 ˚C until used. Plasma samples were thawed on ice for RNA extraction. RNA was isolated from 200 μL of plasma, using the miRNeasy RNA isolation kit (Qiagen, Hilden, Germany) following the manufacturer's instructions. The concentration and purity of isolated RNA were evaluated by NanoDrop 1000 (Nanodrop, Wilmingtion, Delaware, USA) using 1 μl of RNA based on the absorbance measurements at wavelengths of 260 and 280 nm. Samples with 260/280 ratios of ~2.0 is generally accepted as "pure" for RNA. Then, 2 μl of RNA were reverse-transcribed to cDNA using

TaqMan™ Advanced miR cDNA Synthesis Kit, Catalog Number A25576). (Applied Biosystems, Foster City, CA, USA).

Quantitative real-time PCR (RT-qPCR) assay was carried out with a 1.5–mL microcentrifuge from 5 μL of 1/10 diluted cDNA (plasma samples) using 10 μL TaqMan Fast Advanced Master Mix, 1 μL TaqMan Advanced miRNA Assays (Applied Biosystems, Foster City, CA, USA) and 4 μL RNase-free water. The total volume of the mixture product for PCR was 20 μl. The incubation of the mixture product was carried out at 95˚C for 20 seconds followed by 40 cycles of Denature at 95˚C for 1s and Anneal / Extend at 60˚C for 20 seconds.

Real-time quantitative PCR was performed using the Quantistudio 12Kflex Real-Time PCR System (Applied Biosystems, Foster City, CA, USA). Samples were run in duplicate and internal control samples were repeated in every plate to avoid batch effect. The results were analyzed using the RQ manager software (Applied Biosystems). miR-491 was used as normalizer for miR-128, miR-132 and mir-874, while miR-370 was used as normalizer for miR-134, miR-323 and miR-382 according to Sheinerman and co-workers [9]. The expression levels of miR-NAs were calculated using the 2-ΔΔCt method.

## Statistical analysis

Statistical program for social science (SPSS) version 18 for windows SPSS; Inc, Chicago IL was used for data analysis. Chi square test was used for comparing between two qualitative variables. T-test was used for comparing between two means. When data were not normally distributed Mann-Whitney U-tests were used. For comparing between 3 groups, Kruskal Wallis test was used. Two multivariate logistic analyses models were carried out for predicting the convential risk factors and the miRNAs expressions that are significantly associated with MCI among T2DM patients. $P < 0.05$ was considered statistically significant and $p < 0.01$ was considered statistically highly significant. For evaluating the diagnostic performance of each miR-NAs, Receiver-Operating Characteristic (ROC) analysis was carried out for obtaining the area under the curve (AUC) and the corresponding 95% CI. The maximum diagnostic discrimination (MDD) cut-off point was calculated, corresponding to the highest Youden index for each miRNA, followed by detecting the sensitivity, specificity and accuracy for each miRNA were calculated for the identified "cutoff" points.

## Results

The study included 163 T2DM patients aged between 40 and 60 years with mean 53.2 ±5.3 years; 65 (39.9%) of them were males. Non-diabetic normal cognitive 109 individuals were aged between 40–60 years (mean: 51.9 ±4.5 years); 43 (39.4%) of them were males, $P > 0.05$. The percentages of T2DM patients having secondary, university and post graduate education were 46%, 39.9%, and 14.1% compared to 41.3%, 42.2% and 16.5% respectively among non diabetic normal cognitive individuals, $P = 0.716$. Thirty-two (19.6%) of T2DM patients were taking insulin and the rest were taking oral hypoglycemic drugs in the form of sulfonylurea and/or metformin. The ACE III score ranged from 69 to 100 with mean 88.8 ± 6.1. According to ACE III, MCI was present among 59 (36.2%) of the studied T2DM patients. The mean score of ACE III was significantly lower among T2DM patients with MCI (81.5±4.5) compared to those with normal cognition (91.5±3.1), $P < 0.001$. While, the mean score of ACE III was significantly higher among cognitive normal individuals (93.7 ± 3.0) compared the T2DM patients with and without MCI, $P < 0.001$.

## Expressions of miRNAs among non diabetic normal cognitive individuals and T2DM patients with and without MCI

The median miR-132 expression was significantly higher among T2DM patients with MCI compared to those with normal cognition as well as normal non-diabetic cognitive individuals, $P < 0.05$. While the median miR-874 expression was significantly higher among T2DM patients with and without MCI compared to non diabetic normal cognitive individuals, ($P < 0.001$). While, miR-128, miR-134, miR-323, and miR-382 showed no significant difference between the three studied groups, $P > 0.05$ (Table 1).

## Risk factors for MCI among T2DM patients

Socio-demographic and medical data related to MCI among T2DM patients were assessed. Table 2 shows that female sex, low level of education, insulin treatment, presence of sensory peripheral neuropathy are risk factors associated with MCI among T2DM patients.

Table 3 presents logistic regression analysis for predicting the risk of MCI using socioeconomic, laboratory analysis, and miRNA expressions among T2DM patients. It revealed that higher miR- 132 expression, female gender, and below postgraduate education were the significant predicting factors for MCI among T2DM patients ($P < 0.05$).

## Diagnostic performance of miR-128, miR-132 miR-874, miR-134, miR-323, and miR-382 for MCI in T2DM patients

ROC curves for studied miRNAs are presented in Fig 1. The AUC with sensitivity, specificity, and accuracy are presented in Table 4. miR-132 was the only biomarker that significantly differentiates T2DM patients with MCI from those with normal cognition with 72.3% sensitivity, 56.2% specificity, and 63.8% accuracy ($P < 0.05$).

## Discussion

The prevalence of diabetes and MCI are increasing worldwide partially due to the increase in the ageing population and lifestyle choices [12]. Previous studies reported associations of cognitive impairment and/or dementia with T2DM [6,13]

In the present study, MCI was detected among 36.2% of T2DM patients, which is much higher than the percrntage of MCI (7.5%) that was reported among healthy employee [14].

**Table 1. miRNAs expressions among T2DM patients with and without MCI and non-diabetic normal cognitive individuals.**

| Mi- RNA | Non-diabetic normal cognitive individuals | T2DM patients' cognitive function | | P value between the 3 groups |
|---|---|---|---|---|
| | | Normal Cognition | MCI | |
| | Median (Q) | Median (Q) | Median (Q) | |
| miR- 128 | 0.9 (0.3:2.4) | 0.8 (0.3:1.8) | 1.2 (0.5:2.1) | 0.434 |
| miR-132 | 1.2 (0.5:2.9) [αb] | 1.1 (0.4:2.9) | 2.0 (0.7:5.0) [αb] | 0.036 |
| miR-874 | 1.3 (0.4:3.2) | 5.0 (1.6:6.3) [ac] | 5.0 (2.4:6.5) [bc] | < 0.001 |
| miR-134 | 0.6 (0.1:1.8) | 0.8 (0.2:1.9) | 0.6 (0.2:2.0) | 0.227 |
| miR-323 | 0.4 (0.1:1.6) | 0.5 (0.1:1.5) | 0.3 (0.1: 2.1) | 0.993 |
| miR-382 | 0.3 (0.1:1.0) | 0.5 (0.1:1.1) | 0.5 (0.2:1.6) | 0.108 |

**Q** 1st: 3rd quartile.

[αb] Significant difference between T2DM with MCI compared to diabetics and non diabetics with normal cognition, $P < 0.05$.

[bc] Significant difference between T2DM with MCI compared to non-diabetics normal cognition, $P < 0.05$.

[ac] Significant difference between normal cognition with T2DM compared to non-diabetics, $P < 0.05$.

**Table 2. Risk factors for MCI among T2DM patients.**

| Variable | Total | Cognitive function | | Odds Ratio (CI 95%) |
|---|---|---|---|---|
| | n = 163 | MCI (n = 59) | Normal (n = 104) | |
| | | n (%) | n (%) | |
| **Age in years** | | | | |
| 40–49 | 38 | 13 (34.2) | 25 (65.8) | Ⓡ |
| 50–60 | 125 | 46 (36.8) | 79 (63.2) | 1.1 (0.5–2.4) |
| **Gender** | | | | |
| Males | 65 | 16 (24.6) | 49 (75.4) | Ⓡ |
| Females | 98 | 43 (43.9) | 55 (56.1) | 0.4 (0.2–0.8)* |
| **Education** | | | | |
| Secondary | 75 | 37 (49.3) | 38 (50.7) | 21.4 (2.7–167.2)** |
| University | 65 | 21 (32.3) | 44 (67.7) | 10.5 (1.3–83.2) |
| Master/MD | 23 | 1 (4.3) | 22 (95.7) | Ⓡ |
| **Smoking** | | | | |
| Non smokers | 127 | 46 (36.2) | 81 (63.8) | Ⓡ |
| Smokers | 36 | 13 (36.1) | 23 (63.9) | 1.0 (0.5–2.2) |
| **Obesity** | | | | |
| Obese | 100 | 38 (38.0) | 62 (62.0) | 1.2 (0.6–2.4) |
| Non obese | 63 | 21 (33.3) | 42 (66.7) | Ⓡ |
| **Family history of Alzheimer** | | | | |
| Yes | 17 | 6 (35.3) | 11 (64.7) | 1.0 (0.34–2.7) |
| No | 146 | 53 (36.3) | 93 (63.7) | Ⓡ |
| **Hypertension** | | | | |
| Yes | 78 | 29 (37.2) | 49 (62.8) | 1.1 (0.6–2.1) |
| No | 85 | 30 (35.3) | 55 (64.7) | Ⓡ |
| **Duration of diabetes** | | | | |
| ≥ 10 years | 68 | 25 (36.8) | 43 (63.2) | 1.0 (0.5–2.0) |
| < 10 years | 95 | 34 (35.8) | 61 (64.2) | Ⓡ |
| **Treatment received** | | | | |
| Insulin | 32 | 12 (37.5) | 20 (56.3) | 1.1 (0.5–2.4) |
| Oral hypoglycemic | 131 | 47 (35.9) | 84 (74.4) | |
| **Regular DM medication** | | | | |
| Compliant | 124 | 44 (35.5) | 80 (64.5) | Ⓡ |
| Not compliant | 39 | 15 (38.5) | 24 (61.5) | 1.0 (0.4–1.9) |
| **Hypoglycemia** | | | | |
| Yes | 84 | 31 (36.9) | 53 (63.1) | 1.1 (0.6–2.0) |
| No | 79 | 28 (35.4) | 51 (64.6) | Ⓡ |
| **PN** | | | | |
| Yes | 105 | 39 (37.1) | 66 (62.9) | 1.1 (0.5–2.2) |
| No | 58 | 20 (34.5) | 38 (65.5) | Ⓡ |
| **HbA1c%** (mean±SD) | 163 | 7.7±1.8 | 8.1±1.8 | 0.171 |
| **T.C mg/dl** (mean±SD) | 163 | 203.5± 45.5 | 201.4 ±45.8 | 0.777 |
| **Triglycerides mg/dl** (mean±SD) | 163 | 134.8±83.8 | 134.1±67.0 | 0.949 |
| **HDL mg/dl** (mean±SD) | 163 | 43.6±14.5 | 40.7±10.7 | 0.149 |
| **LDL mg/dl** (mean±SD) | 163 | 133.9±34.8 | 135.5±39.8 | 0.801 |

*P <0.05

** P <0.01, CI: Confidence Interval, Ⓡ: reference group, MCI: mild cognitive impairment, PN: peripheral neuropathy, TC: total cholesterol, HDL: high density lipoprotein, LDL: low density lipoprotein.

**Table 3. Logistic regression analysis for predicting the risk of MCI among T2DM patients.**

| Variable | Adjusted Odds Ratio (AOR) | 95% CI | P value |
|---|---|---|---|
| **Gender** | | | |
| Males | Ⓡ | | |
| Females | 2.1 | 1.0–4.3 | 0.05 |
| **Education** | | | |
| Postgraduate | Ⓡ | | |
| University education | 9.5 | 1.2–75.6 | 0.034 |
| Non-university education | 19.4 | 2.5–152.8 | 0.005 |
| **miR-132** | 1.17 | 1.0–1.3 | 0.012 |
| **Constant** | 0.383 | | < 0.001 |

CI: confidence interval; Ⓡ: reference group.

Variables entered in model 1: age, gender, education, DM treatment, DM duration, hypoglycemia, hypertension, and in model 2: miR128, miR132, miR874, miR134, miR323, miR382.

The association of cognitive impairment with T2DM might be related to insulin resistance as insulin receptors are widely distributed throughout the brain. Hyperinsulinemia that precedes or accompanies T2DM increases concentrations of amyloid β through inhibiting its brain clearance. Hyperinsulinemia that precedes or accompanies T2DM increases concentrations of amyloid β peptide (Aβ) through inhibiting its brain clearance. Insulin degrading enzyme (IDE) is responsible for degrading the Aβ as well degrading insulin [15]. IDE is the primary regulator of Aβ in both neurons and glia. Hyperinsulinemia acts as a competitive substrate for this enzyme with Aβ leading to its accumulation forming insoluble plaques [16]. Aβ is one of the main causes of neuronal death during of AD [17]. In the brains of AD and patients with MCI, Aβ have been shown to correlate with rapid cognitive decline [18]. Furthermore, hyper-insulinemia increases plasma and cerebrospinal fluid (CSF) concentrations of interleukin-6 and tumor necrosis factor alpha. All these mechanisms induce neuronal loss, amyloid beta plaques, and neurofibrillary tangles [19].

miRNAs have a significant role in the development of the CNS and in the etiology of many brain pathologies. They have regulatory role in learning and memory function and have an effect on cognition in the normal and diseased brain [20]. MCI and early AD are characterized by destruction of neurite and synapse [21]. miR-132 is widely distributed in the brain and plays a role in regulating neuronal differentiation, maturation and functioning, in adition it participates in axon growth, neural migration and plasticity [22]. Pichler and co-workers [23] reported early and great involvement of miR-132 and miR-212 in the pathogenesis of AD and further stressing on the primarily neuronal origin of these miRNAs. In the present study, The median miR-132 expression was significantly higher among T2DM patients with MCI compared to those with normal cognition as well as normal non-diabetic cognitive individuals, P< 0.05. Logistic regression analysis showed that higher miR-132 expression was a significant predicting risk factor for MCI among T2DM patients, with AOR 1.2 (95% CI 1.0–1.3). It differentiates significantly T2DM patients with MCI from those with normal cognition with 72.3% sensitivity, 56.2% specificity, and 63.8% accuracy. Similarly, Xie and co-workers [24] reported significant upregulation of miR-132 in MCI patients' serum compared with control group. A systematic review done by Wu and co-workers [25] stated that out of the 8 MCI studies, only one miRNA, plasma miR-132, was consistently upregulated in three independent studies. Of the studies that reported diagnostic accuracy data, the majority of miRNA panels and

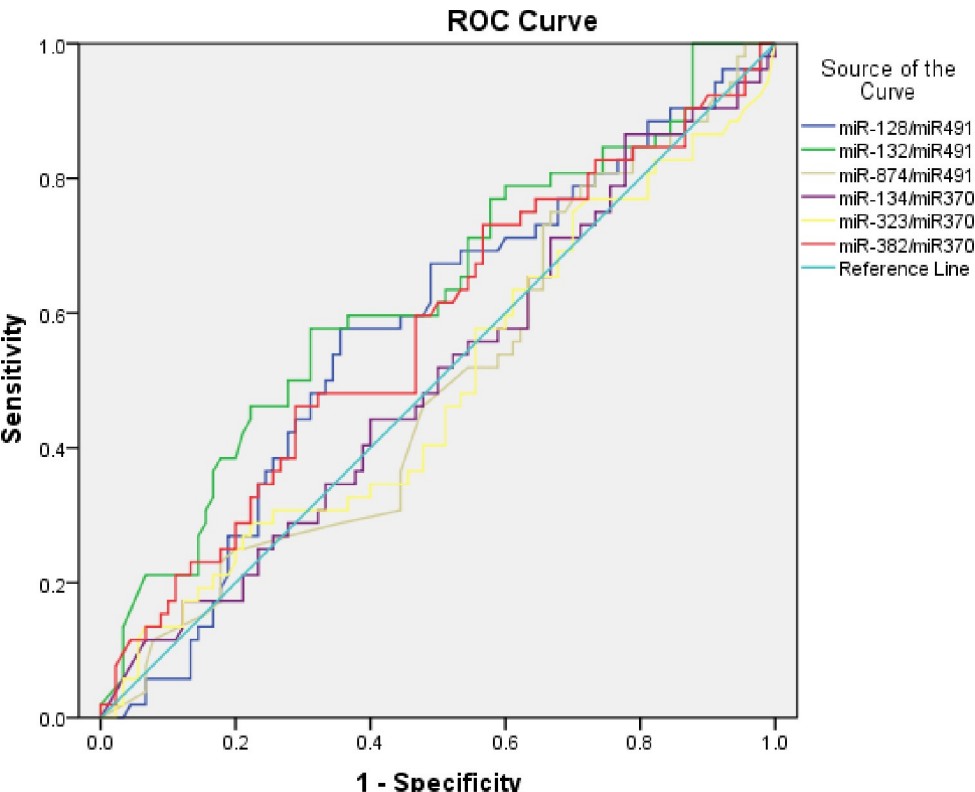

**Fig 1. Receiver-Operating Characteristic (ROC) curve of miR128, miR132, miR874, miR134, miR323, miR382 to evaluate their diagnostic value for MCI among T2DM patients.**

individual miRNAs had a sensitivity and specificity greater than 75%. On the other hand, the knockout of miR-132 in mice was reported to impair learning and memory [26].

El Fatimy and co-workers [27] suggested that miR-132 is a master regulator of neuronal health and miR-132 supplementation could be of therapeutic benefit for the treatment of Tau-associated neurodegenerative disorders. They demonstrated that miR-132 protects primary mouse and human wild-type neurons and more vulnerable Tau-mutant neurons against Aβ and glutamate excitotoxicity. It reduces the level of total, phosphorylated, acetylated, and cleaved forms of Tau implicated in tauopathies, enhances neurite elongation and branching, and decreases neuronal death. Bekris and co-workers [28] assessed five miRNAs including miR-132 in CSF, plasma (obtained during life), cerebellum, and hippocampus (obtained at

**Table 4. Diagnostic performance of studied miRNAs for detecting MCI among T2DM patients.**

| miRNAs | AUC | Sensitivity | Specificity | Accuracy | P-value |
|---|---|---|---|---|---|
| miR-128 | 0.571 (CI: 0.473–0.669) | 61.70% | 61.70% | 61.90% | 0.159 |
| miR-132 | 0.627 (CI: 0.531–0.724) | 72.30% | 56.20% | 63.80% | 0.012* |
| miR-874 | 0.495 (CI: 0.396–0.593) | 82.60% | 24.70% | 45.70% | 0.916 |
| miR-134 | 0.505 (CI: 0.406–0.604) | 53.30% | 50.00% | 51.20% | 0.917 |
| miR-323 | 0.488 (CI: 0.387–0.589) | 61.90% | 50.20% | 54.70% | 0.816 |
| miR-382 | 0.573 (CI: 0.474–0.672) | 72.10% | 42.50% | 53.40% | 0.149 |

*P <0.05; CI: confidence interval.

autopsy). They reported that expressions of these miRNAs in the CSF did not correlate with their expressions in the plasma. This may suggest differences in the expressions of these miRNAs according to the studied tissue type. A study done on transgenic mice showed enhance spatial learning with slight over-expression of hippocampal miR-132. While, with more than three folds overexpression of miR-132 impairs learning. They concluded that miR-132 acting as a key activity-dependent regulator of cognition, which means that the miR-132 expression must be sustained within certain range to ensure normal learning and memory formation [29]. Another study reported an initial increase of miR-132 levels was detected during early AD Braak stages I–II in the human prefrontal cortex, which contrasts with the significant decrease seen at more advanced stages of the disease [30]. Hansen and co-workers [29] found that cognitive capacity is tightly regulated by miR-132 and that physiological levels of miR-132 enhance cognition, whereas supra-physiological levels of the miRNA lead to cognitive deficits. Moreover, overexpression (~3-folds) of miR-132 leads to profound cognitive deficits. Together, these data indicate that miR- 132 plays a key role in shaping cognitive capacity.

miR-128 was reported to be overexpressed in AD patients [31]. In the present study, plasma miR-128 expression was found insignificantly higher in T2DM patients with MCI compared to T2DM patients and non diabetic individuals with normal cognitive functions. This may be explained by the high score of ACE III among T2DM with MCI (mean:81.5) which could indicates that MCI was not severe or was at early stage. Moreover, MÜller and co-workers [32] reported increase expression of miR-128 in the hippocampus in an intermediate stages of AD patients and to be downregulated in late stages of AD patients. In cell cultures, inhibition of miR-128 decreases Aβ-mediated cytotoxicity through inactivation of the NF-κB pathway [31].

miR-874 is involved in axonogenesis, neurotransmitter secretion, dendrite morphogenesis, synaptogenesis, synaptic transmission, and synaptic vesicle exocytosis [33]. Interestingly, in the present study the median miR-874 was overexpressed in T2DM patients with or without MCI compared to non diabetic individuals (P< 0.001), which may indicate its potential role in T2DM. miR-874 was found to be related to fat deposition through promoting insulin synthesis [34]. Moreover, miR-874 was found to be significantly correlated with BMI z-score among overweight/low-grade obesity children and adolescents, but not correlated with the levels of glucose, glycated hemoglobin, total cholesterol and HDL [35]. Additionally, miR-874 is transcriptionally controlled by FoxO3a, which is implicated in insulin/insulin-like growth factor signaling with its effect on gene expression, usually in the opposite direction as insulin and IGF-1 [36]. Similar to the current study, miR-874 expression was not found to be associated with MCI among healthy employee [14].

The current study revealed no significant difference in the median expression of miR-134 between T2DM patients with and without MCI and/or controls, P > 0.05. While Sheinerman and co-workers [9] reported higher levels of the miR-134 in MCI plasma patients compared to controls with 86% sensitivity and 82% specificity. miR-134, is a brain-specific miRNA that has been shown to be upregulated in AD patient brain samples [37]. miR-134 is localized to the synapto-dendritic compartment of rat hippocampal neurons and inhibits neurite growth and local protein synthesis at the synapses [38]. Gao and co-workers [39] reported that miR-134 mediates late long term potentiation (LTP) and synaptic plasticity through the Sirtuin1-cAMP-response-element binding protein (CREB) and brain-derived neurotrophic factor (BDNF) pathway in the hippocampus. miR-134 post-transcriptionally regulates the expression of CREB and BDNF in AD cases [40]. CREB and BDNF are two important plasticity-related proteins that are involved in synaptic plasticity and memory formation [41].

miR-323 is enriched in human pancreatic β compared to α-cells [42] and is considered as diagnostic and prognostic biomarker of beta-cell dysfunction [43]. Moreover, it is a biomarker in inflammatory and immune responses and was found to be associated with the pathogenesis

of many diseases as Alzheimer's disease, rheumatoid arthritis, and ectopic pregnancy [44]. In the current study, median miR-323 expression in T2DM patients with and without MCI was slightly higher compared to normal cognitive controls (P >0.05). Similarly, Kumar and co-workers [45], by using RT-qPCR miRNA assay, they could not confirm the differential expression of plasma miR-323 between AD patients and healthy controls.

miR-382 is present in synaptoneurosomes of cortex, hippocampus and cerebellum [46] and was detected in the brain of some neurological disorders [47]. The current study revealed that there was no significant difference of median expression of miR-382 between the two T2DM patient groups and/or controls, P > 0.05. Rani and co-workers [48] stated that level of miR-382 increased with increasing age but not related to MoCA cognitive test score. Wang and co-workers [49] reported negative correlation between expression of miR-382 in grey matter and the density of early AD lesions subtypes (diffuse plaques, neuritic plaques, and neurofibrillary tangles). While, Sheinerman and co-workers [9] reported higher levels of the miR-134 in MCI plasma patients compared to controls.

A recent systematic review revealed that several miRNAs showed an ability to differentiate between MCI from healthy controls and MCI from AD patients, with modest to high sensitivity and specificity. However, the results of these studies were not always consistent and no consensus has yet been reached. Variations among studies could be due to different reasons including sample size which is widely variable across different studies, different criteria for selection of study participants, differences to define cases and controls, variation of technical procedures used, differences in miRNAs selection and in statistical analysis used [50].

In the current study, logistic analysis revealed that the significant conventional risk factors for predicting MCI among T2DM patients were female gender (AOR 2.1: 95% CI 1.0–4.3) and those with secondary and university education compared to postgraduates (AOR 19.4: 95% CI 2.5–152.8 and 9.5: 95% CI 1.2–75.6, respectively). The association of lower level of education with MCI among T2DM patients may be due to lack of brain stimulation acquired from continuous education upgrading. The higher prevalence of MCI among females may be related to their lower education level. It was reported that participants with normal cognition had significantly higher educational level than those with MCI, with reduced risk of developing MCI and AD [51,52]. Several studies reported that longer duration of T2DM is associated with cognitive decline [53,54]. However, the present study did not detect such association, this might be due to delayed diagnosis of T2DM as most of the studied patients discovered the presence of diabetes accidently.

Some limitations in the current study deserve consideration. Baseline cognitive function scores were not available, which could have helped in comparing the effect of duration on the cognitive function over a time period. In addition, there are other brain-enriched miRNAs which were not assessed in the current study and may be useful as possible biomarkers in the future, computed tomography of the brain was not available and a group of non diabetic patients with MCI were not included to know if the studid miRANs are specific for MCI with T2DM or not.

In conclusion, MCI affects nearly one-third of adult patients with T2DM. A significantly over expression of miR-132 was detected among T2DM with MCI compared to those with normal cognition with sensitivity and specificity < 75%. Other studied miRNAs had lower sensitivity and specificity for detecting MCI among T2DM patients. Therefore, additional studies and extra work are recommended to illuminate their roles as potential biomarkers for MCI.

## Supporting information

**S1 Dataset.**
(XLSX)

## Acknowledgments

The authors gratefully acknowledge the studied participants for their enrollment in the project.

## Author Contributions

**Conceptualization:** Iman I. Salama, Samia M. Sami.

**Formal analysis:** Iman I. Salama, Ghada A. Abdellatif, Amira Mohsen.

**Funding acquisition:** Iman I. Salama.

**Investigation:** Ghada A. Abdellatif, Amira Mohsen, Hanaa Rasmy, Solaf Ahmed Kamel, Mona Hamed Ibrahim.

**Methodology:** Iman I. Salama, Samia M. Sami, Mona Mostafa, Walaa A. Fouad, Hala M. Raslan.

**Project administration:** Samia M. Sami.

**Supervision:** Iman I. Salama, Samia M. Sami, Walaa A. Fouad.

**Validation:** Mona Mostafa, Hala M. Raslan.

**Writing – original draft:** Ghada A. Abdellatif, Amira Mohsen.

**Writing – review & editing:** Iman I. Salama, Samia M. Sami, Ghada A. Abdellatif, Amira Mohsen, Hala M. Raslan.

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
