## [Decision Letter · Decision Letter 0]

1 Jun 2020

PONE-D-20-12036

Plasma microRNAs biomarkers in mild cognitive impairment among patients with type 2 diabetes mellitus

PLOS ONE

Dear Dr. Salama,

Thank you for submitting your manuscript to PLOS ONE. After careful consideration by 2 Reviewers and an Academic Editor, all of the critiques of both Reviewers must be addressed in detail in a revision to determine publication status. If you are prepared to undertake the work required, I would be pleased to reconsider my decision, but revision of the original submission without directly addressing the critiques of the two Reviewers does not guarantee acceptance for publication in PLOS ONE. If the authors do not feel that the queries can be addressed, please consider submitting to another publication medium. A revised submission will be sent out for re-review. The authors are urged to have the manuscript given a hard copyedit for syntax and grammar.

We look forward to receiving your revised manuscript.

Kind regards,

Stephen D. Ginsberg, Ph.D.

Section Editor

PLOS ONE

4. Please include a caption for each figure.

**Comments to the Author**

1. Is the manuscript technically sound, and do the data support the conclusions?

Reviewer #1: Yes

Reviewer #2: Yes

2. Has the statistical analysis been performed appropriately and rigorously? 

Reviewer #1: Yes

Reviewer #2: Yes

3. Have the authors made all data underlying the findings in their manuscript fully available?

Reviewer #1: Yes

Reviewer #2: Yes

4. Is the manuscript presented in an intelligible fashion and written in standard English?

Reviewer #1: Yes

Reviewer #2: Yes

5. Review Comments to the Author

Reviewer #1: In this study, Iman I Salama et al reports that plasma expression of miR-132 was significantly higher in T2DM patients with MCI compared to those without MCI and to normal cognitive healthy subjects.

The cognitive impairment associated with diabetes is a world-wide problem. The article is meaningful because there are no reliable biomarkers for the patients of MCI with T2DM. However, there are several issues that need to be addressed.

1) In the study, the cognitive functions of normal cognitive healthy subjects and T2DM were assessed by the ACE III test, and the score less than 88 was diagnosed as MCI. Authors mentioned that the ACE III score of T2DM patients ranged from 69 to 100, however each score of T2DM groups with and without MCI was not shown. Authors should show the ACE III score of each group of T2DM patients as well as those of cognitive normal subjects.

2) How 163 participants were diagnosed as T2DM? Were they diagnosed by HbA1c and/or blood glucose level? The enrollment criteria of T2DM should be mentioned. In the paper, authors state that 32 of T2DM patients were taking insulin and the rest of patients (n=131) were taking oral hypoglycemic drugs (described in lines 153-154). However, the number of patients received treatments of insulin or oral hypoglycemic was 32 and 117, respectively as shown in Table 2. Please explain this discrepancy.

3) In the paper, the age, sex and education were matched between cognitive normal healthy subjects and T2DM patients (described in lines 82-83). Although the results of age and sex were mentioned in lines 150-152, authors have not shown whether the education level was matched. As shown in Table 2 and 3, the education level seems to be an important factor for the prediction of MCI. Thus, the comparison of education level between control and T2DM patients should be demonstrated.

4) Among the expression of miRNAs including miR-128, miR-132, miR-874, miR-134, miR-323 and miR-382, the expression of miR-132 was significantly higher in T2DM patients with MCI compared to those without MCI. Authors discuss the involvement of miR-128 and miR-874 in addition to miR-132, however there were no implications about the expression of miR-134, miR-323 and miR-382. Please discuss about these miR-134, miR-323 and miR-382 expressions.

5) Throughout the paper, there were many typographical errors such as “miRAs” in line 295. Please correct these wrong notations.

Reviewer #2: The manuscript PONE-D-20-12036 by Salama and colleagues explores the role of several plasma circulating miRNAs to assess their potential as a biomarkers of Mild Cognitive Impairment (MCI), among a population of Type 2 Diabetic patients. The authors focused on the analysis of previous miRNAs known to be associated with MCI. Their results show a significant correlation with high levels of circulating plasma miR-132 in T2D patients with MCI compared to patients without MCI or cognitive healthy individuals. Overall, the work presented by Salama and colleagues is of interest, well written and consistent. However, I recommend a few changes that would improve the quality of the manuscript.

Main points

1. The authors stated that total RNA extracted from plasma was measured with Nanodrop spectrophotometer. Could the authors indicate the amount of plasma RNA from plasma used for retro-transcription to perform miRNA detection? Also, I wonder if the authors were able to analyze the quality and concentration of small RNA by Bioanalyzer or by another alternative method.

2. While explaining the association with T2D with cognitive impairment the authors stated: “Hyperinsulinemia that precedes or accompanies T2DM increases concentrations of amyloid � through inhibiting its brain clearance.” (Line 211-212). Although a reader from the field will probably interpret this sentence easily, the authors might want to make this clear here by adding that amyloid � peptide is one of the main causes of neuronal death during of AD, which is known to be strongly associated with T2DM.

3. Finally, could the authors discuss the possible reason behind the increase in miR-132 in plasma and its potential source? This will maybe clarify the contradictory effect of high levels of miR-132 on MCI, since it is described in the literature that miR-132 exerts a number of beneficial neurolophysiological effects both in human and in mice.

6. PLOS authors have the option to publish the peer review history of their article (what does this mean?). If published, this will include your full peer review and any attached files.

**Do you want your identity to be public for this peer review?** For information about this choice, including consent withdrawal, please see our Privacy Policy.

Reviewer #1: No

Reviewer #2: No

---

## [Author Response · Author response to Decision Letter 0]

27 Jun 2020

Reply to the Reviewer Comments on the manuscript PONE-D-20-12036

We want to express our appreciation to the reviewers for their valuable comments and remarks which added a lot to the manuscript. 

All corrections mentioned below are presented in the revised and in the track changed manuscript files. However, to indicate the site of the corrections done, lines mentioned here are presented in the track changed manuscript file. 

Reviewer #1: In this study, Iman I Salama et al reports that plasma expression of miR-132 was significantly higher in T2DM patients with MCI compared to those without MCI and to normal cognitive healthy subjects. 

The cognitive impairment associated with diabetes is a world-wide problem. The article is meaningful because there are no reliable biomarkers for the patients of MCI with T2DM. However, there are several issues that need to be addressed.

1) In the study, the cognitive functions of normal cognitive healthy subjects and T2DM were assessed by the ACE III test, and the score less than 88 was diagnosed as MCI. Authors mentioned that the ACE III score of T2DM patients ranged from 69 to 100, however each score of T2DM groups with and without MCI was not shown. Authors should show the ACE III score of each group of T2DM patients as well as those of cognitive normal subjects.

Thank you Professor for this valuable comment. We added the following paragraph from line 168 to line 172

"The mean score of ACE III was significantly lower among T2DM patients with MCI (81.5±4.5) compared to those with normal cognition (91.5±3.1), P < 0.001. While, the mean score of ACE III was significantly higher among cognitive normal individuals (93.7 ± 3.0) compared the T2DM patients with and without MCI, P < 0.001".

2) How 163 participants were diagnosed as T2DM? Were they diagnosed by HbA1c and/or blood glucose level? The enrollment criteria of T2DM should be mentioned. In the paper, authors state that 32 of T2DM patients were taking insulin and the rest of patients (n=131) were taking oral hypoglycemic drugs (described in lines 153-154). However, the number of patients received treatments of insulin or oral hypoglycemic was 32 and 117, respectively as shown in Table 2. Please explain this discrepancy.

a) Thank you Professor for this valuable comment. T2DM patients were already diagnosed as T2DM and were receiving anti-diabetics drugs, either insulin or oral hypoglycemic drugs. HbA1c was assessed for detecting the relation between HbA1c level and cognitive function. 

We added and rephrased the following paragraph from line 77 to line 82

"Patients were already diagnosed as T2DM and were receiving anti-diabetics drugs, either insulin or oral hypoglycemic drugs. All patients were literate and were able to complete the tests of cognitive function. T2DM patients with history of head trauma, stroke, transient ischemic attack, brain tumor, epilepsy, psychiatric disease, cardiac or liver failure, history of thyroid disease, visual or hearing disabilities, and acute or chronic infection were excluded." 

b) Yes, Professor you are right. This was a typographical error. Already, the total number T2DM patients for this raw in table 2 was 84 + 47=131 and not 117 as wrongly written. We corrected it to 131. 

3) In the paper, the age, sex and education were matched between cognitive normal healthy subjects and T2DM patients (described in lines 82-83). Although the results of age and sex were mentioned in lines 150-152, authors have not shown whether the education level was matched. As shown in Table 2 and 3, the education level seems to be an important factor for the prediction of MCI. Thus, the comparison of education level between control and T2DM patients should be demonstrated.

Yes Professor, you are right. We added the following paragraph from line 162 to line 165.

"The percentages of T2DM patients having secondary, university and post graduate education were 46%, 39.9%, and 14.1% compared to 41.3%, 42.2% and 16.5% respectively among non-diabetic normal cognitive individuals, P= 0.716." 

4) Among the expression of miRNAs including miR-128, miR-132, miR-874, miR-134, miR-323 and miR-382, the expression of miR-132 was significantly higher in T2DM patients with MCI compared to those without MCI. Authors discuss the involvement of miR-128 and miR-874 in addition to miR-132, however there were no implications about the expression of miR-134, miR-323 and miR-382. Please discuss about these miR-134, miR-323 and miR-382 expressions.

Thank you Professor for this valuable comment. We added the following paragraphs to discuss the mentioned miRNAs and the references were merged with the original manuscript references. We added the following paragraphs from line 301 to line 329

"The current study revealed no significant difference in the median expression of miR-134 between T2DM patients with and without MCI and/or controls, P > 0.05. While Sheinerman and co-workers (9) reported higher levels of the miR-134 in MCI plasma patients compared to controls with 86% sensitivity and 82% specificity. miR‐134, is a brain‐specific miRNA that has been shown to be upregulated in AD patient brain samples (37). miR-134 is localized to the synapto-dendritic compartment of rat hippocampal neurons and inhibits neurite growth and local protein synthesis at the synapses (38). Gao and co-workers (39) reported that miR-134 mediates late long term potentiation (LTP) and synaptic plasticity through the Sirtuin1- cAMP-response-element binding protein (CREB) and brain-derived neurotrophic factor (BDNF) pathway in the hippocampus. miR‐134 post‐transcriptionally regulates the expression of CREB and BDNF in AD cases (40). CREB and BDNF are two important plasticity-related proteins that are involved in synaptic plasticity and memory formation (41).

miR-323 is enriched in human pancreatic β compared to α-cells (42) and is considered as diagnostic and prognostic biomarker of beta-cell dysfunction (43). Moreover, it is a biomarker in inflammatory and immune responses and was found to be associated with the pathogenesis of many diseases as Alzheimer's disease, rheumatoid arthritis, and ectopic pregnancy (44). In the current study, median miR-323 expression in T2DM patients with and without MCI was slightly higher compared to normal cognitive controls (P >0.05). Similarly, Kumar and co-workers (45), by using RT-qPCR miRNA assay, they could not confirm the differential expression of plasma miR-323 between AD patients and healthy controls.

miR-382 is present in synaptoneurosomes of cortex, hippocampus and cerebellum (46) and was detected in the brain of some neurological disorders (47). The current study revealed that there was no significant difference of median expression of miR-382 between the two T2DM patient groups and/or controls, P > 0.05. Rani and co-workers (48) stated that level of miR-382 increased with increasing age but not related to MoCA cognitive test score. Wang and co-workers (49) reported negative correlation between expression of miR-382 in grey matter and the density of early AD lesions subtypes (diffuse plaques, neuritic plaques, and neurofibrillary tangles). While, Sheinerman and co-workers (9) reported higher levels of the miR-134 in MCI plasma patients compared to controls."

 5) Throughout the paper, there were many typographical errors such as “miRAs” in line 295. Please correct these wrong notations.

Thank you Professor, you are right. The manuscript has been thoroughly revised and edited.

Reviewer #2: The manuscript PONE-D-20-12036 by Salama and colleagues explores the role of several plasma circulating miRNAs to assess their potential as a biomarkers of Mild Cognitive Impairment (MCI), among a population of Type 2 Diabetic patients. The authors focused on the analysis of previous miRNAs known to be associated with MCI. Their results show a significant correlation with high levels of circulating plasma miR-132 in T2D patients with MCI compared to patients without MCI or cognitive healthy individuals. Overall, the work presented by Salama and colleagues is of interest, well written and consistent. However, I recommend a few changes that would improve the quality of the manuscript. 

1. The authors stated that total RNA extracted from plasma was measured with Nanodrop spectrophotometer. Could the authors indicate the amount of plasma RNA from plasma used for retro-transcription to perform miRNA detection? Also, I wonder if the authors were able to analyze the quality and concentration of small RNA by Bioanalyzer or by another alternative method.

a) Yes Professor, 2 µl of RNA were reverse-transcribed to cDNA

b) In the current study we used NanoDrop to assess concentration and purity of isolated RNA. While, miRNA expressions was assessed using Quantitative real-time PCR assay according to the manufacture instructions of the TaqMan™ Advanced miR cDNA Synthesis Kit, Catalog Number A25576). (Applied Biosystems, Foster City, CA, USA). We did not use Bioanalyzer or any further alternative method. 

We rephrasing the following paragraph from line 126 to line 131

"The concentration and purity of isolated RNA were evaluated by NanoDrop 1000 (Nanodrop, Wilmingtion, Delaware, USA) using 1 µl of RNA based on the absorbance measurements at wavelengths of 260 and 280 nm. Samples with 260/280 ratios of ~2.0 is generally accepted as “pure” for RNA. Then, 2 µl of RNA were reverse-transcribed to cDNA using TaqMan™ Advanced miR cDNA Synthesis Kit, Catalog Number A25576). (Applied Biosystems, Foster City, CA, USA).

2. While explaining the association with T2D with cognitive impairment the authors stated: “Hyperinsulinemia that precedes or accompanies T2DM increases concentrations of amyloid � through inhibiting its brain clearance.” (Line 211-212). Although a reader from the field will probably interpret this sentence easily, the authors might want to make this clear here by adding that amyloid � peptide is one of the main causes of neuronal death during of AD, which is known to be strongly associated with T2DM.

Thank you Professor for this valuable comment. We added the following paragraph from line 227 to line 235

"Hyperinsulinemia that precedes or accompanies T2DM increases concentrations of amyloid β through inhibiting its brain clearance. Hyperinsulinemia that precedes or accompanies T2DM increases concentrations of amyloid β peptide (Aβ) through inhibiting its brain clearance. Insulin degrading enzyme (IDE) is responsible for degrading the Aβ as well degrading insulin (15). IDE is the primary regulator of Aβ in both neurons and glia. Hyperinsulinemia acts as a competitive substrate for this enzyme with Aβ leading to its accumulation forming insoluble plaques (16). Aβ is one of the main causes of neuronal death during of AD (17). In the brains of AD and patients with MCI, Aβ have been shown to correlate with rapid cognitive decline (18)."

3. Finally, could the authors discuss the possible reason behind the increase in miR-132 in plasma and its potential source? This will maybe clarify the contradictory effect of high levels of miR-132 on MCI, since it is described in the literature that miR-132 exerts a number of beneficial neurolophysiological effects both in human and in mice.

Yes professor, Thank you for this valuable comment. We added the following two paragraphs:

from line 244 to line 246

"Pichler and co-workers (23) reported early and great involvement of miR-132 and miR-212 in the pathogenesis of AD and further stressing on the primarily neuronal origin of these miRNAs." 

from line 273 to line 280.

"An initial increase of miR-132 levels was detected during early AD Braak stages I–II in the human prefrontal cortex, which contrasts with the significant decrease seen at more advanced stages of the disease (30). Hansen and co-workers (29) found that cognitive capacity is tightly regulated by miR-132 and that physiological levels of miR-132 enhance cognition, whereas supra-physiological levels of the miRNA lead to cognitive deficits. Moreover, overexpression (~3-folds) of miR-132 leads to profound cognitive deficits. Together, these data indicate that miR- 132 plays a key role in shaping cognitive capacity."

---

## [Decision Letter · Decision Letter 1]

8 Jul 2020

Plasma microRNAs biomarkers in mild cognitive impairment among patients with type 2 diabetes mellitus

PONE-D-20-12036R1

Dear Dr. Salama,

We’re pleased to inform you that your manuscript has been judged scientifically suitable for publication and will be formally accepted for publication once it meets all outstanding technical requirements.

Kind regards,

Stephen D. Ginsberg, Ph.D.

Section Editor

PLOS ONE

**Comments to the Author**

1. If the authors have adequately addressed your comments raised in a previous round of review and you feel that this manuscript is now acceptable for publication, you may indicate that here to bypass the “Comments to the Author” section, enter your conflict of interest statement in the “Confidential to Editor” section, and submit your "Accept" recommendation.

Reviewer #1: All comments have been addressed

Reviewer #2: All comments have been addressed

2. Is the manuscript technically sound, and do the data support the conclusions?

Reviewer #1: Yes

Reviewer #2: Yes

3. Has the statistical analysis been performed appropriately and rigorously? 

Reviewer #1: Yes

Reviewer #2: I Don't Know

4. Have the authors made all data underlying the findings in their manuscript fully available?

Reviewer #1: Yes

Reviewer #2: Yes

5. Is the manuscript presented in an intelligible fashion and written in standard English?

Reviewer #1: Yes

Reviewer #2: Yes

6. Review Comments to the Author

Reviewer #1: (No Response)

Reviewer #2: (No Response)

7. PLOS authors have the option to publish the peer review history of their article (what does this mean?). If published, this will include your full peer review and any attached files.

Reviewer #1: No

Reviewer #2: No

---

## [Editor Report · Acceptance letter]

14 Jul 2020

PONE-D-20-12036R1 

Plasma microRNAs biomarkers in mild cognitive impairment among patients with type 2 diabetes mellitus 

Dear Dr. Salama:

I'm pleased to inform you that your manuscript has been deemed suitable for publication in PLOS ONE. Congratulations! Your manuscript is now with our production department. 

Kind regards, 

on behalf of

Dr. Stephen D. Ginsberg 

Section Editor

PLOS ONE